# Dried Destoned Virgin Olive Pomace: A Promising New By-Product from Pomace Extraction Process

**DOI:** 10.3390/molecules26144337

**Published:** 2021-07-17

**Authors:** Cinzia Benincasa, Massimiliano Pellegrino, Lucia Veltri, Salvatore Claps, Carmelo Fallara, Enzo Perri

**Affiliations:** 1CREA Research Centre for Olive, Fruit and Citrus Crops, C.da Li Rocchi, 87036 Rende, CS, Italy; cinzia.benincasa@crea.gov.it (C.B.); massimiliano.pellegrino@crea.gov.it (M.P.); 2Chemistry Department, University of Calabria, Cubo 12C, 87036 Rende, CS, Italy; lucia.veltri@unical.it; 3CREA Research Centre for Animal Production and Aquaculture, S.S. 7 Via Appia, 85051 Bella Muro, PZ, Italy; salvatore.claps@crea.gov.it; 4Olearia S.O.D., Via del Progresso, 64023 Mosciano Sant’Angelo, TE, Italy; oleariasod@libero.it

**Keywords:** olive, phenols, olive pomace, olive by-products, functional foods, zootechnical products, mass spectrometry

## Abstract

At present the olive oil industry produces large amounts of secondary products once considered waste or by-products. In this paper, we present, for the first time, a new interesting olive by-product named “dried destoned virgin olive pomace” (DDVOP), produced by the pomace oil industry. The production of DDVOP is possible thanks to the use of a new system that differs from the traditional ones by having the dryer set at a lower temperature value, 350 °C instead of 550 °C, and by avoiding the solvent extraction phase. In order to evaluate if DDVOP may be suitable as a new innovative feeding integrator for animal feed, its chemical characteristics were investigated. Results demonstrated that DDVOP is a good source of raw protein and precious fiber; that it is consistent in total phenols (6156 mg/kg); rich in oleic (72.29%), linoleic (8.37%) acids and tocopherols (8.80 mg/kg). A feeding trial was, therefore, carried out on sheep with the scope of investigating the influence of the diet on the quality of milk obtained from sheep fed with DDVOP-enriched feed. The resulting milk was enriched in polyunsaturated (0.21%) and unsaturated (2.42%) fatty acids; and had increased levels of phenols (10.35 mg/kg) and tocopherols (1.03 mg/kg).

## 1. Introduction

Secondary products from the olive oil industry are the most abundant agro-industrial by-products in the Mediterranean area [1]. Depending on the extraction technology, the main possible secondary products are seven: (i) an aqueous liquid (olive mill wastewater, OMW) and a solid waste (olive pomace, OP) from traditional and three-phase systems; (ii) a semisolid waste (olive cake, OC) from two-phase systems; (iii) a semisolid destoned waste (paté olive cake, POC) and fragments of olive stone (FOS), from new two-phase decanters, and a de-oiled pomace from the pomace oil industry. OP, as a by-product, is characterized by oily residues, lignocellulosic matrix, high-fiber and low-protein content, saturated and unsaturated fatty acids, tocopherols and phenolic compounds [2,3,4,5]. Phenols are very strong antioxidants and in many experimental studies, they have demonstrated a wide spectrum of biological and pharmacological activities, beyond their antioxidant properties [6,7,8,9,10,11]. They protect the body from free radicals which are constantly formed during physiological processes and responsible for the occurrence of oxidative stress which is related to different physiological and pathological processes such as aging, cancer, cardiovascular diseases, and diabetes [10,11,12,13,14]. A large number of epidemiological studies indicate the existence of an inverse relationship between dietary intake of foods rich in antioxidants and the incidence of many nutrition related diseases [15]. Recently, new research shows that olive-derived phenols can exert pharmacological effects in the prevention of inflammation and oxidative stress [9]. Olive phenols can also be used by the food industry as natural food additives with antioxidant and pharmacological properties and for extending the shelf life of food [16]. Three-phase extraction systems, which are still widely used in Italian olive oil mills, require the addition of large amounts of water (up to 50 L/100 kg olive paste), while two-phase systems involve a lower amount of olive mill wastewater (OMW) volumes but an increased concentration in organic matter [17]. Both types of waste are a significant source of environmental pollution as they are characterized by high chemical oxygen demand (COD), unpleasant color and odor, acidic pH, high concentration of salt and phenolic compounds. OMW inhibits seed germination and early plant growth [18], alters soil characteristics [19] and creates reducing conditions, affecting microbial diversity in soil [20]. In earlier studies, OMW toxicity was attributed to low molecular weight phenolics, in particular, monomeric phenolic compounds [21,22] and phenols, such as *p*-coumaric and ferulic acids, were proved to affect the physiology of both prokaryotic and eukaryotic organisms [23]. OMW has been, also, reported to decrease the phosphorylation efficiency of mitochondria [24]. On the other hand, phenols from OMW can be used to control and inactivate plant and human pathogens [25], in the inactivation of pathogenic bacteria and their toxins [26] and in the suppression against common weeds and nematodes [27]. However, still now olive mill secondary products are not easily reusable. Therefore, at present, OMW may be used mainly as a water source, while OP may be exploited for the production of pits, compost, soil conditioners, natural fertilizers, biomass for biogas production [28], or for the production of pomace oil through chemical extraction and refining of the residual oil (10–12% of OP). Considering that most phenolic compounds present in the olive are mainly polar, only a small amount is solubilized in the oil during the malaxing operation, olive mill secondary products represent a rich and cheap source of these bioactive compounds (95–98%) [15]. Then, a suitable use of these phenols could positively affect the economic status of olive companies and reduce the negative impact of olive by-products on the environment. Phenols extracted could be applied as natural antioxidants and potentially useful for different types of biological as well as pharmacological applications [6,15,16]; in cosmetics as anti-aging and anti-wrinkle products, as well as in nutrition as food integrators [15,29].

Numerous studies have shown that phenols play a direct role in animal welfare and indirect on humans when transferred to milk and meat [30,31,32]. Nowadays, the use of “functional” foods containing substances of natural origin rather than synthetic antibiotics is promoted for the welfare of animals and the healthiness of food derivatives [31]. In fact, one of the most important challenges for the livestock industry consists of increasing productivity without an increase in the production costs, minimizing the environmental impact and maximizing the animal welfare [33]. The use of olive cake in animal feeding in livestock production is of recent experimentation, but the studies carried out demonstrated the very interesting potential for use in relation to improving the nutritional quality of meat [34,35,36,37,38,39,40,41]. Other studies, also in ruminantes, and concerning the production of milk and cheeses [30,41], showed a significant effect of dietary supplementation with fresh destoned olive pomace on saturated, monounsaturated, linoleic acid, *n*-3 and *n*-6 polyunsaturated fatty acids (PUFAs) content and nutritional index. Consequently, the trend at the European and national levels is to increase the use of bioactive molecules in animal feeding to reduce the employment of synthetical chemical compounds and antibiotics.

In this paper we present, for the first time, a new interesting olive industries secondary product named “dried destoned virgin olive pomace” (DDVOP), produced by the olive pomace industry by a simple modification in the flux diagram and technical parameters of traditional technology used to extract pomace oil from virgin olive pomace. In particular, the present study aimed to characterize and evaluate the chemical characteristics of this new product and investigate if DDVOP is suitable as a new feeding integrator for animal feed.

To evaluate this application strategy, a feeding trial was carried out on sheep in order to investigate the influence of the diet on the quality of milk derived from animals fed with conventional and with DDVOP-enriched feed. The objective was to monitor and compare the specific transfer and enrichment of selected bioactive constituents of DDVOP-enriched feed such as fatty acids, phenolic compounds and tocopherols in milk as well as subsequently its quality for secondary human consumption.

## 2. Materials and Methods

### 2.1. Dried Destoned Virgin Olive Pomace (DDVOP)

DDVOP (Figure 1) was obtained from drying fresh virgin olive pomace in a SIAM brand rotary dryer, equipped with a combustion pre-oven powered by exhausted pomace produced on site. The main feature of this dryer that distinguishes it from traditional ones is its reduced air temperature (Figure 2). Fresh olive pomaces were obtained in 2019 by mechanical extraction from three phase oil mills situated in Cosenza, a southern province of Italy. Before drying, virgin olive pomace was stored at a temperature between 25 °C and 30 °C in large heaps.

### 2.2. DDVOP Phenols Analysis

The extraction of phenols from DDVOP was carried out on 10 g of DDVOP using, as extracting solvent, a solution of methanol/water (*v*/*v* 80:20) (20 mL). To facilitate the migration of phenols from the matrix to the solution, the mixture was kept under stirring in an ultrasonic bath in the dark at 6 °C for 15 min. After this period, centrifugation at 5000 rpm/min (centrifugal force of 1677× *g*) for 25 min allowed the separation of the phases and the recovery of the supernatant. The analyses were performed by HPLC-MS/MS. In particular, an HPLC 1200 series instrument (Agilent Technologies, Santa Clara, CA, USA) equipped with an Eclipse XDB-C8-A HPLC column (5 μm particle size, 150 mm length and 4.6 mm i.d.) was used for chromatographic separation. An MSD Sciex Applied Biosystem API 4000 Q-Trap mass spectrometer was used to detect the phenolic compounds. Each standard compound was monitored by multiple reaction monitoring (MRM) mode which scans, on the third quadrupole, the main fragments of the deprotonated molecular ion [M − H]^−^ produced in the first quadrupole. The standards used for the quantitative analyses were purchased from Sigma–Aldrich (Riedel-de Haën, Laborchemikalien, Seelze) and Extrasynthese (Nord B.P 62 69,726 Genay Cedex, France) and were: catechol (Cat), caffeic acid (Caf), vanillin (Van), vanillic acid (Vco), *p*-cumaric acid (*p*-Cum), o-cumaric acid (*o*-Cum), ferulic acid (Fer), apigenin (Ap), apigenin-7-*O*-glucoside (Ap7), diosmetin (Dio), hydroxytyrosol (HyTyr), tyrosol (Tyr), oleuropein (Olp), luteolin (Lut), verbascoside (Ver), luteolin-7-*O*-glucoside (Lu7), luteolin-4-*O*-glucoside (Lu4), rutin (Rut) and syringic acid (Syr). The MRM transitions monitored for the assay were: 109 → 91 for Cat; 179 → 135 for Caf; 151 → 136 for Van; 167 → 108 for Vco; 163 → 119 for *o*-Cum and *p*-Cum; 193 → 134 for Fer; 269→ 117 for Ap; 4312 → 268 for Ap7; 299 → 284 for Dio; 153 → 123 for HyTyr; 137 → 137 for Tyr; 539 → 307 and 539 → 275 for Olp; 285 → 133 for Lut; 623 → 161 and 623 → 461 for Ver; 447 → 285 for Lut7 and Lut4; 609 → 301 for Rut; 197 → 121 for Syr. The LC-MS experimental conditions for all the analytes under investigation were as follows: ionspray voltage (IS) −4600 V; curtain gas 23 psi; temperature 400 °C; ion source gas (1) 35 psi; ion source gas (2) 40 psi; collision gas thickness (CAD) medium. The instrument parameters such as entrance potential (EP), declustering potential (DP), collision energy (CE) and cell exit potential (CXP) were optimized for each transition monitored (Table 1). The chromatographic separation was achieved at a flow rate of 300 µL/min with an injection volume of 10 µL. A binary mobile phase made up of 0.1% aqueous formic acid (A) and methanol (B) was gradient programmed to increase B from 5% to 100% in 15 min, hold for 5 min and ramp down to original composition (95% A and 5% B) in five min. Quantitative analyses were performed by external calibration curves built using a least-squares linear regression analysis. For this purpose, standard stock solutions of the phenolic compounds of interest were dissolved in methanol and further diluted with water/0.1% formic acid to obtain calibration standards at concentrations in the range between 100 and 2000 μg/mL. All the solvents used were LC/MS grade; aqueous solutions were prepared using ultrapure water (Millipore, Bedford, MA, USA). The correlation coefficients of the calibration curve ranged between 0.9997 and 0.9999. To evaluate the amount of the oleuropein and ligstroside derivatives, a quantification based on oleuropein glycoside peak areas was deployed. Unfortunately, a direct quantification by means of calibration curves was not possible for these two compound families because they are not commercially available.

### 2.3. DDVOP Tocopherols Analysis

Tocopherols determination was achieved according to a method previously adopted by Benincasa and co-workers for olive oil analysis and adapted for the new matrix [7,42]. In summary, 10 g of DDVOP were extracted with hexane and made up to volume (10 mL). The resulting solution was filtered by means of Whatman® membrane filters (PFTE 0.2 µm pore size, 25 mm diam., Sigma-Aldrich Riedel-de Haen, Laborchemikalien, Seelze) and injected (20 µL) into an HPLC system (Agilent Technologies, Santa Clara, CA, USA) equipped with a Zorbax NH_2_ column (25 cm × 4.6 mm i.d., 5 µm particle size, Agilent Technologies, Santa Clara, CA, USA) using an isocratic mobile phase of hexane/ethyl acetate (80:20 *v*/*v*). The flow rate was 1.5 mL/min and the detector, a fluorescence spectrophotometer with a programmed wavelength (295 and 325 nm). The standards used for the quantitative analyses were purchased from Sigma-Aldrich (Riedel-de Haën, Laborchemikalien, Seelze, Germany). Hexane was LC grade; aqueous solutions were prepared using ultrapure water (Millipore, Bedford, MA, USA). Quantitative analyses were performed by external calibration curves built using a least-squares linear regression analysis. For this purpose, standard stock solutions of α-tocopherol (α-toc), β-tocopherol (β-toc), γ-tocopherol (γ-toc) and δ-tocopherol (δ-toc) were dissolved in hexane and further diluted to obtain calibration standards at concentrations in the range between 10 and 100 μg/mL. The correlation coefficients of the calibration curve ranged between 0.995 and 0.999. The results were expressed in mg of α, β, γ and δ tocopherol per kg of DDVOP. Three independent experiments were carried out to calculate mean and SD.

### 2.4. DDVOP Fatty Acid Analysis

Fatty acid composition was determined as methyl esters (FAME) following the procedures described in the enclosures of the Commission Regulation EEC no. 2568/91. The procedure was slightly modified to adapt it to the new matrix [7,42]. In summary, 1 g of DDVOP was dissolved in 10 mL of hexane and 1 mL of a methanolic solution of KOH (1 N). The resulting solution was shaken vigorously for 5 min. Subsequently, 0.25 mL of the supernatant was taken, deposited in a vial and dissolved in 1.5 mL of hexane. Then, 1 µL hexane solution was injected into a gas chromatograph (Agilent 6890N) equipped with a capillary column SP-2340 (60 m × 0.25 mm i.d., 0.2 µm f.t., Supelco). The separation was carried out with a programmed temperature (110 °C held for 5 min, increased by 3 °C/min to 150 °C and held for 16.33 min, then increased by 4 °C/min to 230 °C and held for 27 min) and an FID detector at 260 °C. The results were expressed in the percentage of chromatographic areas. Three independent experiments were carried out to calculate mean and SD.

### 2.5. Experimental Trial on Sheep and Milk Analyses

A feeding trial was carried out at “Bella” estate, a farm of CREA—Research Centre for Animal Production and Aquaculture, of about 70 ha where about 900 native sheep and goats are grazing. A group of ten sheep (group M2) was fed with enriched food for ten days while another group of ten sheep (group M1) received normal feed. Milk samples were collected before the treatment from both groups in order to have a bulk sample as a control after ten days of the experiment. The animal study was reviewed and approved by the Ethics Committee for animal testing–CES, of the Research Centre for Animal Production and Aquaculture. The extraction of phenols from the milk was carried out according to analytical methods described by Vázquez and co-workers [43]. Tocopherols were separated and quantified by ultra high performance liquid chromatography (HPLC), following the method developed and validated by Moltó-Puigmartì and co-workers [44]. Milk proteins were precipitated with ethanol and tocopherols were extracted with hexane. The hexane extract, dried under nitrogen, was reconstituted in a dichloromethane:methanol (2:1, *v*/*v*) solution. Three independent experiments were carried out to calculate mean and SD. Tocopherols were separated by means of a reversed-phase chromatographic column by using acetonitrile:methanol (60:40, *v*/*v*) as a mobile phase. The detection was carried out by means of a fluorescence detector, using a range of excitation wavelengths between 295 and 325 nm. Quantitative analyses were performed by external calibration curves as already described in Section 2.3. Three independent experiments were carried out to calculate mean and SD. Fatty acid analyses were determined as methyl esters (FAME) following the procedures described in Section 2.4. Three independent experiments were carried out to calculate mean and SD.

### 2.6. Chemical Analyses of DDVOP for Animal Feed

The chemical analyses of DDVOP were determined according to the method reported by Goering and Van Soest [45].

### 2.7. Statistical Analysis

The statistical treatment was performed by using the statistics program STATGRAPHICS Plus Version 5.1 (Statistical Graphics Corporation, Professional Edition—copyright 1994–2001). The identification of differences among groups was evaluated by one-way analysis of variance (ANOVA) and Tukey’s post hoc test for multiple comparisons with statistical significance at a 95% confidence level (*p* < 0.05).

## 3. Results and Discussion

Obtaining this new promising olive industry secondary product, DDVOP, was possible by a simple modification in the flux diagram and technical parameters of traditional technology used to extract pomace oil from virgin olive pomace. The new device differs from the traditional ones by having the dryer set at a lower temperature value and by avoiding the solvent extraction phase (Figure 2). In particular, the olive virgin pomace is added from the higher end into a rotary dryer, a cylinder that is slightly inclined from the horizontal direction, mainly composed of a rotating body, a lifting plate, a transmission device, a supporting device and a sealing ring. The high-temperature heat source and the material parallelly flow or counter-flow into the cylinder. As the cylinder rotates, the material runs to the lower end. The lifting board on the inner wall of the cylinder lifts up and drops down the material so that the contact surface of the material and the air flow is increased to increase the drying rate and promote the advancement of the material. The dried product (Figure 1) exits the dryer from the lower end.

The technical characteristics concerning the drying process and the differences between the traditional and the new process in terms of oven and dryer air temperature are shown in Table 2. The new conditions, with the oven outlet air temperature of 350 °C instead of 550 °C, dryer outlet air temperature of 70 °C instead of 80 °C, allow a significant decrease in dried olive pomace temperature, from 70 °C to 40 °C, while leading to a product of the same final moisture (8%). The innovative flow leading to DDVOP production differs from the traditional ones, also, for having, immediately after the drying device, an aspirator instead of an extractor. To produce DDVOP, in fact, no solvent extraction takes place and no pomace oil, to be distilled, is obtained. The aspiration process, instead, allows the removal of the stone making DDVOP suitable, after further toxicological evaluation, as a supplement amendment for animal feed. From the data obtained, DDVOP resulted in a high content of organic substance and a good quantity of raw protein and precious fiber, composed of neutralized fiber (59.9 g/100 g dry matter (DM)) and clean acid fiber (39.7 g/100 g DM), which makes it very suitable as a supplement in animal feed (Table 3). In fact, fibers, that cannot be used as an energy source in monogastric organisms having a single-chambered stomach (one stomach) are instead a very good energy source in ruminants as they can degrade them by means of rumen microbes. DDVOP contains a low lignin content, lower than the quantity present in the pitted pomace and comparable to that of a pomace obtained from pitted olives. This low quantity makes the digestibility of fibers in ruminants even better; also, lignin, described as a potent in vitro source of antioxidants and adsorbers of hydrophobic carcinogens in the whole intestine, has been claimed as carriers of phenolics throughout the gastrointestinal tract (Mudgil, 2017; Sato et al., 2011). DDVOP was found to be very consistent in total phenols (6156 mg/kg) and rich in oleic (72.29%) and linoleic (8.37%) acids [42]. These values reflect the typical composition of olive oil (Table 4). DDVOP, also, contains α-tocopherol (2.69 mg/kg), β-tocopherol (4.32 mg/kg), δ-tocopherol (0.29 mg/kg) and γ-tocotrienol (0.48 mg/kg) (Table 5). Tocopherols are particularly important functional components in foods. They have vitamin E properties and display antioxidant activity, which protect the body tissues against the damaging effects caused by the free radicals that result from many normal metabolic functions. The data concerning the analysis of single phenols were very interesting. Considering that oleuropein and ligstroside are two of the major components of olive leaves and olive drupes, their concentration was also found to be high in DDVOP (30.22; 149.37 and 433.72 mg/kg of oleuropein, oleuropein derivatives and ligstroside derivatives, respectively). These phenolic compounds are well known for their blood pressure-lowering effects, derived health benefits and their anti-inflammatory and antioxidant properties against atherosclerosis, diabetes, cancer, neurodegenerative diseases and even arthritis [46,47]. DDVOP is very rich in substituted phenols such as hydroxytyrosol (339.27 mg/kg), tyrosol (512.25 mg/kg), catechol (220.32 mg/kg), vanillin (20.36 mg/kg), *p*-cumaric acid (73.25 mg/kg), syringic acid (0.35 mg/kg), *o*-cumaric acid (25.44 mg/kg), caffeic acid (83.85 mg/kg), ferulic acid (3.88 mg/kg) and vanillic acid (89.54 mg/kg) (Table 6). Among them, hydroxytyrosol and tyrosol are known for protecting low-density lipoproteins and consequently reducing cardiovascular disease risk [48]. DDVOP was, also, found to be rich in flavonoids: apigenin (17.87 mg/kg); apigenin-7-*O*-glucoside (0.13 mg/kg); luteolin (1248 mg/kg); luteolin-7-*O*-glucoside (43.66 mg/kg); luteolin-4-*O*-glucoside (26.89 mg/kg); diosmetin (27.84 mg/kg); rutin (51.48 mg/kg) and verbascoside (541.08 mg/kg) [48,49]. Among them, luteolin possesses relevant biological properties, such as antioxidant, anti-inflammatory, anti-microbial and cardio-tonic activity, ability to scavenge free radicals and to inhibit low-density lipoprotein oxidation [49,50]; luteolin-7-*O*-glucoside and luteolin-4-*O*-glucoside are responsible for the color of the drupes.

As the results obtained clearly demonstrated that DDVOP was suitable for a direct use in formulated feed, thanks, also, to its ideal moisture content (11 g/100 g DM), DDVOP was included in a specially formulated feed in a dose of 6%. The specially formulated feed was tested by constituting two experimental groups (homogeneous for live weight, lactation stage and body condition score) of Sardinian lactating sheep. In addition to the basic hay, both groups formed of ten sheep each, the control group (M1) and the experimental one (M2), were fed with a design specifically for lactating sheep, weather-resistant, protein block with fat, vitamins and minerals added to balance nutrient deficiencies in fair quality forages. Animal feed for M2 was enriched with 6% of DDVOP. The diet of the two groups were, then, isoenergetic and isoproteic. Milk samples were collected before starting the experimental trial and after ten days. The results obtained are tabulated in Table 7. The data showed a significant effect (*p* < 0.05) for the fatty acid profile, phenols and tocopherols. The use of concentrate containing DDVOP induced, in fact, an increase in PUFA (0.179 and 0.211% respectively) and UFA (2.161 and 2.423%, respectively), with possible beneficial effects on human health, while inducing a decrease in SFA (5.126 and 4.755%, respectively). Moreover, the use of concentrate containing DDVOP induced a highly significant increase in phenols (4.316 and 10.346 mg/kg, respectively) and an increase in tocopherols (0.891 and 1.034 mg/kg, respectively).

## 4. Conclusions

The technical characteristics concerning the drying process, with the oven outlet air temperature of 350 °C instead of 550 °C, dryer outlet air temperature of 70 °C instead of 80 °C, allows a significant decrease in dried olive pomace temperature, from 70 °C to 40 °C, while leading to a product of the same final moisture (8%). Lowering the temperature by 30 °C allowed to obtain a DDVOP rich in raw protein, precious fiber, phenols, tocopherols, oleic and linoleic acids, hence, making this new product suitable as a new feeding integrator for animal feed. DDVOP, included in a specially formulated feed in a dose of 6%, gave a significant effect (*p* < 0.05) on milk obtained from sheep fed with DDVOP-enriched feed in terms of phenols, tocopherols, polyunsaturated and unsaturated fatty acids with possible beneficial effects on human health. The production of DDVOP could represent a fine and cheap source of bioactive compounds that positively affect the economic status of olive companies while reducing the negative impact of olive by-products on the environment.

## Figures and Tables

**Figure 1 molecules-26-04337-f001:**
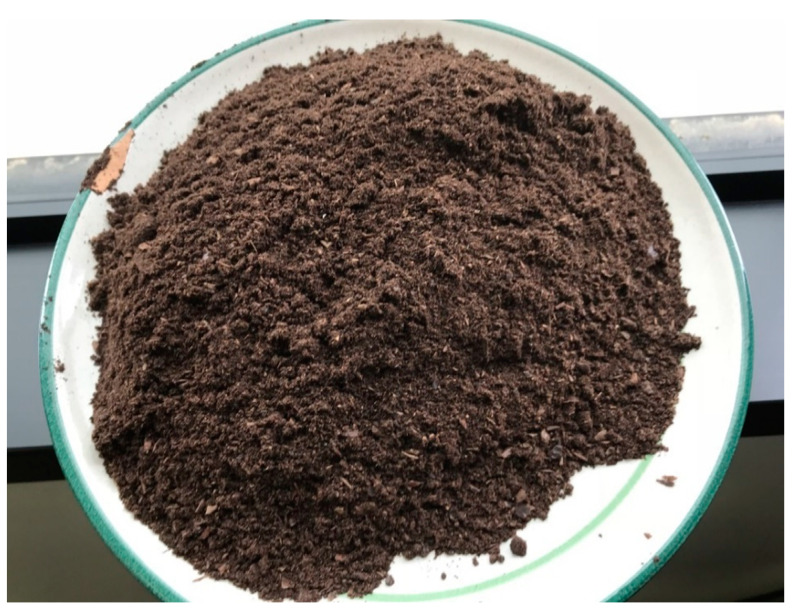
Dried destoned virgin olive pomace (DDVOP).

**Figure 2 molecules-26-04337-f002:**
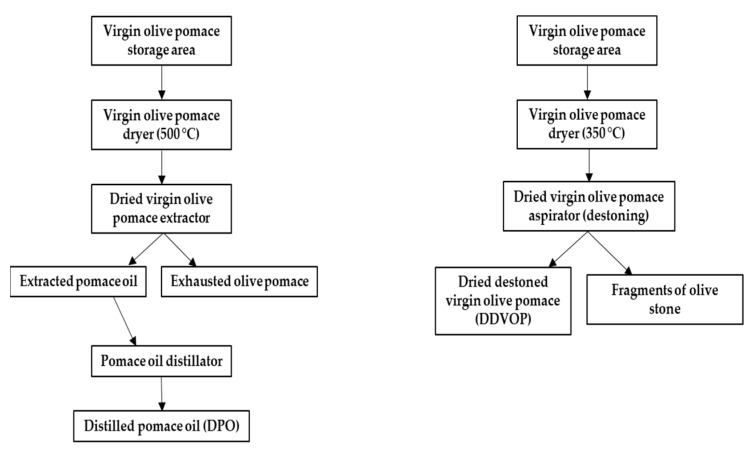
Simplified olive pomace flow sheet: (**left**) traditional flow; (**right**) innovative flow leading to the dried destoned virgin olive pomace (DDVOP) production.

**Table 1 molecules-26-04337-t001:** LC-MS/MS analysis parameters for the phenolic compounds under investigation including molecular ion [M − H]^−^ monitored on the first quadrupole; main fragment ion selected on the third quadrupole (*m*/*z*); declustering potential (DP); entrance potential (EP); collision energy (CE) and cell exit potential (CXP).

Phenolic Compound	[M − H]^−^	Main Fragment Ion	DP	EP	CE	CXP
	(*m*/*z*)	(*m*/*z*)	(V)	(V)	(V)	(V)
Catechol	108.8	90.9	−83	−11	−22	−5
Caffeic acid	178.8	135.1	−60	−11	−20	−6
Vanillin	150.9	135.9	−61	−10	−25	−6
Vanillic acid	166.8	108.1	−33	−10	−25	−4
*p*-Coumaric acid	162.8	119.0	−31	−7	−17	−5
*o*-Coumaric acid	162.8	119.0	−27	−7	−17	−5
Ferulic acid	192.9	134.1	−35	−10	−21	−5
Apigenin	269.0	117.0	−96	−11	−44	−5
Apigenin-7-*O*-Glucoside	431.0	267.1	−90	−10	−40	−6
Diosmetin	299.0	284.1	−107	−11	−35	−13
Hydroxytyrosol	152.9	123.0	−80	−11	−22	−5
Tyrosol	136.9	119.1	−60	−8	−20	−6
Oleuropein	539.3	377.1	−65	−10	−20	−5
Luteolin	284.5	133.2	−104	−5	−37	−6
Verbascoside	623.3	161.1	−83	−11	−39	−15
Luteolin-7-*O*-Glucoside	447.1	285.1	−82	−11	−35	−7
Luteolin-4-*O*-Glucoside	447.1	285.1	−60	−10	−32	−15
Rutin	609.2	301.0	−116	−11	−50	−15
Syringic acid	196.8	120.9	−70	−11	−25	−6

**Table 2 molecules-26-04337-t002:** Technical data concerning the drying process.

Operational Spry Drying Conditions	Traditional Process	New Process
Oven outlet air temperature	550 °C	350 °C
Dryer outlet air temperature	80 °C	70 °C
Dried olive pomace temperature	70 °C	40 °C
Residual moisture of dried olive pomace	8%	8%

**Table 3 molecules-26-04337-t003:** Dried destoned virgin olive pomace (DDVOP) chemical characteristics. The data represent the mean values of three replications with their relative standard deviation (RSD). Values are expressed as the percentage of dry matter (g/100 g DM). Phenols are expressed as mg/kg.

Chemical Characteristics	Value
Crude Protein	8.7 ± 0.4
Ether Extract	17.6 ± 3.3
Neutral Detergent Fiber	59.9 ± 1.3
Acid Detergent Fiber	39.7 ± 1.1
Acid Detergent Lignin	14.8 ± 0.3
Ash	7.7 ± 0.4
Total Phenols	6156 ± 187
Moisture	11.2 ± 0.1

**Table 4 molecules-26-04337-t004:** Fatty acids profile (% FAME) of dried destoned virgin olive pomace (DDVOP). The data represent the mean values of three replications with their relative standard deviation (RSD).

Fatty Acids	%
C14:0	0.02 ± 0.00
C16:0	13.49 ± 1.41
C16:1	1.61 ± 0.31
C17:0	0.13 ± 0.02
C17:1	0.31 ± 0.08
C18:0	2.27 ± 0.13
C18:1n9	72.29 ± 2.38
C18:2n6	8.37 ± 0.71
C18:3n3	0.60 ± 0.07
C20:0	0.34 ± 0.06
C20:1n9	0.32 ± 0.08
C22:0	0.11 ± 0.02
C24:0	0.13 ± 0.04

**Table 5 molecules-26-04337-t005:** Tocopherols (mg/kg) of dried destoned virgin olive pomace (DDVOP). The data represent the mean values of three replications with their relative standard deviation (RSD).

Tocopherol	mg/kg
α-tocopherol	2.69 ± 0.11
ß-tocopherol	4.32 ± 0.09
δ-tocopherol	0.29 ± 0.12
γ-tocotrienol	0.48 ± 0.10
Total tocopherols	7.78 ± 0.42

**Table 6 molecules-26-04337-t006:** Phenols content (mg/kg) in dried destoned virgin olive pomace (DDVOP) by liquid chromatography–tandem mass spectrometry (LC-MS/MS).

Phenolic Compound	mg/kg
Catechol	220.32 ± 0.54
Tyrosol	512.25 ± 39.62
Vanillin	20.36 ± 1.89
Hydroxytyrosol	339.27 ± 4.77
p-Cumaric acid	73.25 ± 15.13
Syringic acid	0.35 ± 0.08
*o*-Cumaric acid	25.44 ± 5.66
Caffeic acid	83.85 ± 5.73
Ferulic acid	3.88 ± 0.05
Vanillic acid	89.54 ± 7.93
Apigenin	17.87 ± 5.49
Apigenin-7-*O*-glucoside	0.13 ± 0.01
Luteolin	1248 ± 104
Luteolin-7-*O*-glucoside	43.66 ± 5.28
Luteolin-4-*O*-glucoside	26.89 ± 4.41
Diosmetin	27.84 ± 8.58
Rutin	51.48 ± 7.44
Oleuropein	30.22 ± 1.73
Verbascoside	541.08 ± 23.26
Oleuropein derivatives	149.37 ± 5.20
Ligstroside derivatives	433.72 ± 9.50
Sum of phenols	3795 ± 209

**Table 7 molecules-26-04337-t007:** Effect of dried destoned virgin olive pomace (DDVOP) on the chemical characteristic of sheep milk. The values (mean ± SD) are expressed as a percentage for acid composition and mg/kg for phenols and tochopherols. M1—milk from sheep fed unaltered feed; M2—milk from sheep fed feed enriched with DDVOP; PUFA—polyunsaturated fatty acids; SFA—saturated fatty acids; UFA —unsaturated fatty acids. Different letters indicate statistical differences as per *p* < 0.05.

Group	PUFA	SFA	UFA	Phenols	Tocopherols
M1	0.179 ^b^ ± 0.028	5.126 ^a^ ± 0.477	2.161 ^b^ ± 0.211	4.316 ^b^ ± 0.301	0.891 ^b^ ± 0.094
M2	0.211 ^a^ ± 0.039	4.756 ^b^ ± 0.452	2.423 ^a^ ± 0.282	10,346 ^a^ ± 1.221	1.034 ^a^ ± 0.063

## Data Availability

The data that support the findings of this study are available. Addi-tional information can be requested from the corresponding author upon reasonable request.

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
