# Peer review of "Dried Destoned Virgin Olive Pomace: A Promising New By-Product from Pomace Extraction Process"

_molecules, 2021, doi:10.3390/molecules26144337_

Round 1

Reviewer 1 Report

The paper entitled ‘Dried Destoned Virgin Olive Pomace: a Surprising and Promis-2 ing New By-product from Pomace Extraction Process’ prepared by Benincasa et al. aimed to characterize and evaluate of dried destoned virgin olive pomace. On the whole, the paper is interesting but in this form should not be accepted to publication. The process can be considered after the following changes and improvements:

  1. Abstract: there is too small information about obtained results. Conclusions should be added.
  2. Results and discussion: the part is not readable and should be improve. Authors present obtained results in form of tables but the tables should be more precise discussed. This part include results but there is no significant discussion. There are two figures but there is no added commentary to them.
  3. Conclusions: conclusions are obligatory. Please provide this section.

Author Response

We thank you for the interest given to this paper and for the insightful comments. This helped us to rectify and improve the quality of our manuscript. We have carefully reviewed the comments and revised the manuscript accordingly. Following are the detailed corrections listed point by point.

Reviewer comments:

The paper entitled ‘Dried Destoned Virgin Olive Pomace: a Surprising and Promising New By-product from Pomace Extraction Process’ prepared by Benincasa et al. aimed to characterize and evaluate of dried destoned virgin olive pomace. On the whole, the paper is interesting but in this form should not be accepted to publication. The process can be considered after the following changes and improvements:

  1. Abstract: there is too small information about obtained results. Conclusions should be added.

Abstract has been improved according the referee advice. The new abstract reads now as follows:

“At present the olive oil industry produces large amounts of secondary products once considered waste or by-products. In this paper we present, for the first time, a new interesting olive by-product named “dried destoned virgin olive pomace” (DDVOP), produced by the pomace oil industry. The production of DDVOP was possible thanks to the use of a new system that differs from the traditional ones for having the dryer set at lower temperature value, 350 °C instead of 550 °C, and for avoiding the solvent extraction phase. In order to evaluate if DDVOP may be suitable as a new innovative feeding integrator for animal feed, its chemical characteristics were investigated. Results demonstrated that DDVOP is a good source of raw protein and precious fiber; that it is consistent in total phenols (6156 mg/kg); rich in oleic (72,29 %), linoleic (8.37 %) acids and tocopherols (8,80 mg/kg). A feeding trial was, therefore, carried out on sheep with the scope of investigating the influence of the diet on the quality of milks obtained from sheep fed with DDVOP-enriched feed. Milks resulted enriched in polyunsaturated (0.21%) and unsaturated (2.42%) fatty acids; increased in phenols (10.35 mg/kg) and tocopherols (1.03 mg/kg)”.

Conclusion paragraph has been added (see point 3.)

  1. Results and discussion: the part is not readable and should be improve. Authors present obtained results in form of tables but the tables should be more precise discussed. This part include results but there is no significant discussion. There are two figures but there is no added commentary to them.

Results and discussion have been improved according to the referee advice. The new test now reads as follows:

The obtaining of this new promising olive industries secondary product, DDVOP, was possible by a simple modification in the flux diagram and technical parameters of traditional technology used to extract pomace oil from virgin olive pomace. The new device differs from the traditional ones for having the dryer set at lower temperature value and for avoiding the solvent extraction phase (Figure 2). In particular, the olive virgin pomace is added from the higher end into a rotary dryer, a cylinder that is slightly inclined from the horizontal direction, mainly composed of a rotating body, a lifting plate, a transmission device, a supporting device and a sealing ring. The high-temperature heat source and the material parallelly flow or counter-flow into the cylinder. As the cylinder rotates, the material runs to the lower end. The lifting board on the inner wall of the cylinder lifts up and drops down the material, so that the contact surface of the material and the air flow is increased to increase the drying rate and promote the advancement of the material. The dried product (Figure 1) exits the dryer from the lower end. The technical characteristics concerning the drying process and the differences between the traditional and the new process in terms of oven and dryer air temperature are shown in Table 2. The new conditions, with the oven outlet air temperature of 350 °C instead of 550 °C, dryer outlet air temperature of 70 °C instead of 80 °C, allow a significant decrease in dried olive pomace temperature, from 70 °C to 40 °C, while leading to a product of the same final moisture (8 %). The innovative flow leading to DDVOP production differs from the traditional ones, also, for having, immediately after the drying device, an aspirator instead an extractor. To produce DDVOP, in fact, no solvent extraction takes place and no pomace oil, to be distilled, is obtained. The aspiration process, instead, allows the removal of the stone making DDVOP suitable as supplement amendment for animal feed. From the data obtained, DDVOP resulted to have a high content of organic substance and a good quantity of raw protein and precious fiber, composed of neutralized fiber (59.9 g/100g dry matter (DM)) and clean acid fiber (39.7 g/100g DM), which makes it very suitable as a supplement in zootechnical feeding (Table 3). In fact, fibers, that cannot be used as energy source in simple stomaches are instead a very good energy source in ruminant’s diet as they can degradate them by means of rumen microbes. DDVOP contains a low lignin content, lower than the quantity present in the pitted pomace and comparable to that of a pomace obtained from pitted olives. This low quantity makes the digestibility of fibers in ruminants even better; also, lignin, described as potent in vitro source of antioxidants and adsorbers of hydrophobic carcinogens in the whole intestine, has been claimed as carriers of phenolics throughout the gastrointestinal tract (Mudgil, 2017; Sato et al., 2011). DDVOP was found to be very consistent in total phenols (6156 mg/kg) and rich in oleic (72,29 %) and linoleic (8.37 %) acids [42]. These values reflect the typical composition of an olive oil (Table 4). DDVOP, also, contains α-tocopherol (2,69 mg/kg), ß-tocopherol (4,32 mg/kg), δ-tocopherol (0,29 mg/kg) and γ-tocotrienol (0,48 mg/kg) (Table 5). Tocopherols are particularly important functional components in foods. They have vitamin E properties and display antioxidant activity, which protect the body tissues against the damaging effects caused by the free radicals that result from many normal metabolic functions. The data concerning the analysis of single phenols were very interesting. Considering that oleuropein and ligstroside are two of the major components of olive leaves and olive drupes, their quantity in DDVOP were found to be quite significative (30,22; 149,37 and 433,72 mg/kg of oleuropein, oleuropein derivatives and ligstroside derivatives, respectively). These phenolic compounds are very important for their blood pressure-lowering effects and health benefits ad for their anti-inflammatory and antioxidant properties for fighting atherosclerosis, diabetes, cancer, neurodegenerative diseases, and even arthritis [47, 48]. DDVOP is very rich in substituted phenols such as hydroxytyrosol (339,27 mg/kg), tyrosol (512,25 mg/kg), catecol (220,32 mg/kg), vanillin (20,36 mg/kg), p-Cumaric acid (73,25 mg/kg), syringic acid (0,35 mg/kg), o-Cumaric acid (25,44 mg/kg), caffeic acid (83,85 mg/kg), ferulic acid (3,88 mg/kg) and vanillic acid (89,54 mg/kg) (Table 6). Among them, hydroxytyrosol and tyrosol are important for protecting low-density lipoproteins and consequently reducting cardiovascular disease risk [49]. DDVOP was, also, found rich in flavonoids: apigenin (17,87 mg/kg); apigenin-7-O-glucoside (0,13 mg/kg); luteolin (1248 mg/kg); luteolin-7-O-glucoside (43,66 mg/kg); luteolin-4-O-glucoside (26,89 mg/kg); diosmetin (27,84 mg/kg); rutin (51,48 mg/kg) and verbascoside (541,08 mg/kg) [49, 50]. Among them, luteolin possess important biological properties, such as antioxidant, anti-inflammatory, anti-microbial and cardio-tonic activity, ability to scavenge free radicals and to inhibit low-density lipoprotein oxidation [50, 51]; luteolin-7-O-glucoside and luteolin-4-O-glucoside are responsible for the colour of the drupes. As the results obtained clearly demonstrated that DDVOP was suitable for a direct use in formulated feed thanks, also, to its ideal moisture content (11 g/100g DM), DDVOP was included in a specially formulated feed in a dose of 6%. The specially formulated feed was tested by constituting two experimental groups (homogeneous for live weight, lactation stage and body condition score) of Sardinian lactating sheep. In addition to the basic hay, the witness group (M1) was fed with a company concentrate containing no pomace, while the experimental group (M2) was fed with a concentrate containing 6% of DDVOP. The diet of the two groups were isoenergetic and isoproteic. Milk samples were collected before starting the experimental trial and after ten days. The results obtained are tabulated in Table 7. The data showed a significant effect (P< 0.05) for the fatty acid profile, phenols and tocopherols. The use of concentrate containing DDVOP induced, in fact, an increase in PUFA (0.179 and 0.211% respectively) and UFA (2.161 and 2.423%, respectively), with possible beneficial effect on human health, while induced a decrease content in SFA (5.126 and 4.755%, respectively). Moreover, the use of concentrate containing DDVOP induced a highly significant increase in phenols (4.316 and 10.346 mg/kg, respectively) and an increase in tocopherols (0.891 and 1.034 mg/kg, respectively).

  1. Conclusions: conclusions are obligatory. Please provide this section.

Conclusion paragraph has been added and reads as follows:

“The technical characteristics concerning the drying process, with the oven outlet air temperature of 350 °C instead of 550 °C, dryer outlet air temperature of 70 °C instead of 80 °C, allow a significant decrease in dried olive pomace temperature, from 70 °C to 40 °C, while leading to a product of the same final moisture (8%). Lowering the temperature of 30 °C allowed to obtain a DDVOP rich in raw protein, precious fiber, phenols, tocopherols, oleic and linoleic acids, hence, making this new product suitable as a new feeding integrator for animal feed. DDVOP, included in a specially formulated feed in a dose of 6%, gave a significant effect (P< 0.05) on milks obtained from sheep fed with DDVOP-enriched feed in terms of phenols, tocopherols, polyunsaturated and unsaturated fatty acids with possible beneficial effect on human health. The production of DDVOP could represent a fine and cheap source of bioactive compounds that positively affect economic status of olive companies while reducing the negative impact of olive by-products on the environment”.

Reviewer 2 Report

The current manuscript investigates the possibilities of utilizing a by-product rich in selected secondary plant metabolites from the olive industry for animal feeding. The purpose is clearly praxis driven and enables the valorisation of the olive by products. In the following I will try to indicate some parts where I think corrections are recommended.

Title: I would suggest removing “a surprising”, as I suppose that it was only a matter of time till the potential was detected, esp. as more bio-economic based research is being directed to close up the processing cycles and the trend has been observed by increasing publications in the corresponding fields. Finally, thereafter the title would become more precise and compact:- Dried Destoned Virgin Olive Pomace: a Promising New By-product from Pomace Extraction Process

Abstract: Lines 14-18: Starting “In general ….” This part belongs to introduction part – I would recommend the corresponding transfer giving the reader a more detailed insight of possible by-products produced during the processing of olives.

Further, the content of the abstract can be enhanced by adding some concrete data with regard to the bioactive compounds present and followed during the preparation of dried destoned virgin olive pomace. In the same context, some concrete data from the feeding experiment should be given to highlight the improvements observed.

Line 23: …. as a new innovative feeding integrator for animal feed.

Line 27: … higher in milks derived by sheep… I would recommend writing “….higher in milks from sheep fed with … Principally, it would be better to replace “ derived by” with simply putting “from” or “obtained from” throughout the manuscript.

Introduction: The introduction gives a good and compact resume of the work done and encompasses the main themes needed to be addressed. A more detailed insight of possible by-products would enrich this part. Further, the authors write in abstract following: “Thus, the present study aims to characterize and evaluate if DDVOP, obtained by a simple modification in the flux diagram and technical parameters of traditional technology used to extract pomace oil from virgin olive pomace …..” – it would also be helpful for the reader to explain/highlight this part of the approach and how it fits in the complete technological processing workflow while referring to Figure 2.

Line 43: … oxidants and the incidence of many nutrition related diseases [15]. Recently, new research shows that olive-derived ….

Line 50: Owing to its toxicity, antimicrobial activity and ineffective management, ….  Can you please elaborate more on the toxicity aspects – which compounds are responsible for it and how the current processing step avoids these effects – esp. while animal and/or human consumption is envisaged.

Line 64: In fact, since the phenolic compounds in the olive’s past are mainly … This parts needs to be re-written – perhaps so: In fact, many phenolic compounds present in the olives are primarily characterized by their polar nature, such that only a small amount is……..

Line 89: … chemical characteristics of this new…

Line 90: Please replace here and elsewhere in the manuscript “zootechnical food” by “animal feed”.

Line 90: To evaluate this application strategy, a feeding trial was carried out on sheep in order to investigate the influence of the diet on the quality of milks obtained by animals fed with conventional and with DDVOP-enriched feed. The objective was to monitor and compare the specific transfer and enrichment of selected bioactive constituents of DDVOP-enriched feed such as fatty acids, phenolic compounds and tocopherols in milk as well as to subsequently improve its quality for the secondary human consumption.

Line 95: Starting “The results …..” This sentence is not recommended here and should be removed, further it is also a repetition.

Material and methods

The methods applied are well established protocols in line with the intended objectives – no new method development has been presented. So again here also the innovative input has been kept rather at a low level. Some method details are missing and need to be added to permit reproduction of the data. Even a few aspects to the experimental design need to be clarified, these are listed in the following specific remarks:

Line 101: … Cosenza, a South province of Italy. Please add here this sentence: The processing steps were modified as indicated in Figure 1.

Line 102: …. stored at room temperature in large heaps for some days before its drying was…  Can you be more precise – this would certainly provoke deterioration of the product (via oxidation and microbial growth) – were the changes monitored e.g. loss of phenolic compounds, anti-oxidative potential? Please define the “room temperature with a range (e.g. 25-30°C ?). Please compare Figure 1, the colouring is very dark brown suggesting that extensive oxidation of phenolic compounds has taken place here and in the successive processing steps, which represent future sites for improvement to prevents such losses.

Line 104: …. exhausted pomace produced on site (details are given in Table 1).

Line 106-107: This part needs to be re-written :-  The extraction of phenols was conducted with 10g DDVOP under shaking in an ultrasonic bath in darkness (at room temperature?) for 15 minutes with (xxx mL) of a solution of methanol/water (v/v 80:20). Please give the amount of the solution used and please use SI units where possible throughout the manuscript (e.g. g instead of gr.).

Line 108: … a centrifugation at 5000 rpm/min for 25 min… please give the centrifugation speed with e.g. 5000 x g, which in turn depends on the centrifuge used.

Line 112-138: Following details are missing: Injection volume used, flow rate, gradient conditions, MS conditions used.

Lines 119-127: Please prepare a table for these substances giving parent ion, fragment ion considered and collision energy applied.

 Line 132: Please check the journal guide lines and use corresponding units (ml-1 or mL-1) and be consistent there after throughout in the manuscript.

Line 136: … on oleuropein glycoside peak areas was deployed.

Line 149: Hexane was LC grade;…..

Line 171: The official permission or grant for the animal experiments is missing!

Lines 177 and 191: Why did the authors use different methods as compared to the ones described in the former sections? A comparison is there after difficult, as different parameters apply e.g. fluorescence detection was changed to excitation wavelength of 297nm and an emission wavelength of 328 nm (compare line 147).

Results and discussion:

The results and discussion is poorly presented and needs a polishing in the language issues. Finally, I am also missing a more in-depth discussion to the outcomes.

Line 213-214: DDVOP was produced …  This sentence is not clear, please re-write it.

Line 219: 9.9 % DM? Please give the corresponding units here and elsewhere in the manuscript.

Line 220: Please replace “zootechnical feeding” with animal feed here and elsewhere in the manuscript!

Line 221: …. in simple stomaches are … Be scientifically more precise!

Line 230-231: It should be: 0.29 mg/kg, please replace comma with point here and elsewhere in the manuscript (please see also lines 242-250).

Line 237: … in DDVOP were found to be significantly high. ???

Line 239: These phenolic compounds are well known for their ….

Line 240: … effects derived health benefits and for their anti-inflammatory ….

Line 240 …. properties against atherosclerosis, ….

Line 243: … catechol ...

Line 243-244: p-cumaric/o-cumaric acids – Please do not use capitals and “o” or “p” should be cursive.

Line 246: … tyrosol are known for protecting….

Line 247: …reducing …

Line 247: … was, also, found to be rich …

Line 251: … possess relevant …

Line 254: As the results obtained clearly demonstrated that DDVOP was suitable for a direct use in formulated feed thanks?, ... This is not completely true, since no further toxicological evaluation was done! Please re-write!

Line 259: …. the control group (M1) ?

Line 260: …. with a company concentrate… – What do you mean?

Line 257-260: This is a repetition!

Conclusions are missing!!!!

Author Response

We thank the Reviewer 2 for the interest given to this paper and for the insightful comments. This helped us to rectify and improve the quality of our manuscript. We have carefully reviewed the comments and revised the manuscript accordingly. Following are the detailed corrections listed point by point.

Title

Dried Destoned Virgin Olive Pomace: a Surprising and Promising New By-product from Pomace Extraction Process

was changed in:

“Dried Destoned Virgin Olive Pomace: a Promising New By-product from Pomace Extraction Process”.

Abstract: Line 14-18 are now

In general, depending on the extraction technology, the main possible secondary products are seven: i) an aqueous liquid (olive mill wastewater, OMW) and a solid waste (olive pomace, OP) from traditional and three-phase systems; ii) a semisolid waste (olive cake, OC) from two-phase systems; iii) a semisolid destoned waste (paté olive cake, POC) and fragments of olive stone (FOS), from new two-phase decanters, and a de-oiled pomace from pomace oil industry.

This part has been removed to the Introduction section.

Abstract has been improved according the referee advice. The new abstract reads now as follows:

“At present the olive oil industry produces large amounts of secondary products once considered waste or by-products. In this paper we present, for the first time, a new interesting olive by-product named “dried destoned virgin olive pomace” (DDVOP), produced by the pomace oil industry. The production of DDVOP was possible thanks to the use of a new system that differs from the traditional ones for having the dryer set at lower temperature value, 350 °C instead of 550 °C, and for avoiding the solvent extraction phase. In order to evaluate if DDVOP may be suitable as a new innovative feeding integrator for animal feed, its chemical characteristics were investigated. Results demonstrated that DDVOP is a good source of raw protein and precious fiber; that it is consistent in total phenols (6156 mg/kg); rich in oleic (72,29 %), linoleic (8.37 %) acids and tocopherols (8,80 mg/kg). A feeding trial was, therefore, carried out on sheep with the scope of investigating the influence of the diet on the quality of milks obtained from sheep fed with DDVOP-enriched feed. Milks resulted enriched in polyunsaturated (0.21%) and unsaturated (2.42%) fatty acids; increased in phenols (10.35 mg/kg) and tocopherols (1.03 mg/kg)”.

Line 23

“zootechnical food” and “derived by”

was replaced by “animal feed” and “obtained from” thrughout all the manuscript

Line 27

higher in milks derived by sheep fed with DDVOP-enriched feed

now reads: “..higher in milks from sheep fed with DDVOP-enriched feed”.

Line 43

…antioxidants and the incidence of a lot of diseases

now reads: “…antioxidants and the incidence of many nutrition related diseases”

Line 50

Owing to its toxicity, antimicrobial activity and ineffective management…

The text was elaborated more on the toxicity as the reviewer adviced.

New sentences and references (from 18 to 27) were added as follow:

“Both types of waste are a significant source of environmental pollution as they are characterized by high chemical oxygen demand (COD), unpleasant color and odor, acidic pH, high concentration of salt and phenolic compounds. OMW inhibit seed germination and early plant growth [18], alter soil characteristics [19] and create reducing conditions, affecting microbial diversity in soil [20]. In earlier studies, OMW toxicity was attributed to low molecular weight phenolics, in particular monomeric phenolic compounds [21, 22] and phenols, such as p-coumaric and ferulic acids, were proved to affect the physiology of both prokaryotic and eukaryotic organisms [23]. OMW has been, also, reported to decrease the phosphorylation efficiency of mitochondria [24]. On the other hand, phenols from OMW can be used to control and inactivate plant and human pathogens [25], in the inactivation of pathogenic bacteria and their toxins [26] and in the suppression against common weeds and nematodes [27]. However, still now olive mill secondary products are not easily reusable. Therefore, at present, OMW may be used mainly as water source, while OP may be exploited for the production of pits, compost, soil conditioners, natural fertilizers, biomass for biogas production [28], or for the production of pomace oil through chemical extraction and refining of the residual oil (10-12 % of OP). Considering that most phenolic compounds present in the olive are mainly polar, only a small amount is solubilized in the oil during the malaxing operation, olive mill secondary products represent a rich and cheap source of these bioactive compounds (95-98%) [15]. Then, a suitable use of this phenols could positively affect economic status of olive companies and reduce the negative impact of olive by-products on the environment. Phenols extracted could be applied as natural antioxidants and potentially useful for different types of biological as well as pharmacological applications [6, 15-16]; in cosmetics as anti-aging and anti-wrinkle products, as well as in nutrition as food integrators [15-29].

Line 64

In fact, since the phenolic compounds in the olive’s past are mainly…

now reads: “In fact, many phenolic compounds present in the olive are mainly…”

Line 89

….chemical characteristic of this new….

now reads:

“….chemical characteristics of this new….”

Line 90

“zootechnical food” was replaced by “animal feed”

“To valuate so, a feeding trial was carried out on sheep in order to investigate the influence of the diet on the quality of milks derived by animals fed with conventional feeding stuffs and milks derived by animals fed with DDVOP-enriched feed. Fatty acids, phenolic compounds and tocopherols were found to be higher in milks derived by sheep fed with DDVOP-enriched feed. The results obtained showed differences between the two types of milk: fatty acids, phenolic compounds and tocopherols were found to be higher in milks from sheep fed with DDVOP-enriched feed”

Now reads:

“To valuate this application strategy, a feeding trial was carried out on sheep in order to investigate the influence of the diet on the quality of milks derived by animals fed with conventional and with DDVOP-enriched feed. The objective was to monitor and compare the specific transfer and enrichment of selected bioactive constituents of DDVOP-enriched feed such as fatty acids, phenolic compounds and tocopherols in milks as well as subsequently its quality for the secondary human consumption.

Line 95

The results obtained showed differences between the two types of milk: fatty acids, phenolic compounds and tocopherols were found to be higher in milks from sheep fed with DDVOP-enriched feed.

The sentence was removed.

Line 101 and 102

Fresh olive pomaces were obtained in 2018 and 2019 by mechanical extraction from three phase oil mills situated in Cosenza, a South province of Italy. Virgin olive pomace was stored at room temperature in large heaps for some days before its drying was carried out. The drying of virgin olive pomace was carried out in a SAIM brand rotary dryer, equipped with combustion pre-ovens, powered by exhausted pomace produced on site.

The paragraph has been re-written and now it reads::

“DDVOP (Figure 1) was obtained from drying fresh virgin olive pomace in a SIAM brand rotary dryer, equipped with a combustion pre-oven powered by exhausted pomace produced on site. The main feature of this dryer is the reduced air temperature compared to traditional ones (Figure 2)”. Fresh olive pomaces were obtained in 2019 by mechanical extraction from three phase oil mills situated in Cosenza, a South province of Italy. Before drying, virgin olive pomace was stored at a temperature between 25 and 30 °C in large heaps”.

Line 104

…exhausted pomace produced on site.

The paragraph 2.1. Dried Destoned Virgin Olive Pomace (DDVOP) was re-written as follos:

“DDVOP (Figure 1) was obtained from drying fresh virgin olive pomace in a SIAM brand rotary dryer, equipped with a combustion pre-oven powered by exhausted pomace produced on site. The main feature of this dryer that distinguishes it from traditional ones is its reduced air temperature (Figure 2). Fresh olive pomaces were obtained in 2019 by mechanical extraction from three phase oil mills situated in Cosenza, a South province of Italy. Before drying, virgin olive pomace was stored at a temperature between 25 and 30 °C in large heaps”.

Line 106-107

The extraction of phenols from DDVOP was performed by keeping under shaking in an ultrasonic bath in the darkness for 15 minutes a solution of methanol/water (v/v 80:20) containing 10 gr of DDVOP.

Now it reads:

“The extraction of phenols from DDVOP was carried out on 10 g of DDVOP using, as extracting solvent, a solution of methanol/water (v/v 80:20) (20 mL). To facilitate the migration of phenols from the matrix to the solution, the mixture was kept under stirring in an ultrasonic bath in the dark at 6 °C for 15 min. After this period, a centrifugation at 5000 rpm/min (centrifugal force of 1677 × g) for 25 min allowed the separation of the phases and the recovery of the supernatant”.

Line 108

…a centrifugation at 5000 rpm/min for 25 min…

now reads:

“…a centrifugation at 5000 rpm/min (centrifugal force of 1677 × g) for 25 min allowed the separation of the phases and the recovery of the supernatant”.

Line 112-138

The paragraph 2.2. DDVOP Phenols Analysiswas re-written and it i sas follows:

“The extraction of phenols from DDVOP was carried out on 10 g of DDVOP using, as extracting solvent, a solution of methanol/water (v/v 80:20) (20 mL). To facilitate the migration of phenols from the matrix to the solution, the mixture was kept under stirring in an ultrasonic bath in the dark at 6 °C for 15 min. After this period, a centrifugation at 5000 rpm/min (centrifugal force of 1677 × g) for 25 min allowed the separation of the phases and the recovery of the supernatant. The analyses were performed by HPLC-MS/MS. In particular, an HPLC 1200 series instrument (Agilent Technologies, Santa Clara, California) equipped with an Eclipse XDB-C8-A HPLC column [(5 μm particle size, 150 mm length and 4.6 mm i.d.) was used for chromatographic separation. An MSD Sciex Applied Biosystem API 4000 Q-Trap mass spectrometer was used to detect the phenolic compounds. Each standard compound was monitored by multiple reaction monitoring (MRM) mode which scans, on the third quadrupole, the main fragments of the deprotonated molecular ion [M-H]-1 produced in the first quadrupole. The standards used for the quantitative analyses were purchased from Sigma–Aldrich (Riedel-de Haën, Laborchemikalien, Seelze) and Extrasynthese (Nord B.P 62 69726 Genay Cedex, France) and were: catecol (Cat), caffeic acid (Caf), vanillin (Van), vanillic acid (Vco), p-cumaric acid (p-Cum), o-cumaric acid (o-Cum), ferulic acid (Fer), apigenin (Ap), apigenin-7-O-glucoside (Ap7), diosmetin (Dio), hydroxytyrosol (HyTyr), tyrosol (Tyr), oleuropein (Olp), luteolin (Lut), verbascoside (Ver), luteolin-7-O-glucoside (Lu7), luteolin-4-O-glucoside (Lu4), rutin (Rut) and syringic acid (Syr). The MRM transitions monitored for the assay were: 109 → 91 for Cat; 179 → 135 for Caf; 151 → 136 for Van; 167 → 108 for Vco; 163 → 119 for o-Cum and p-Cum; 193 → 134 for Fer; 269→ 117 for Ap; 4312 → 268 for Ap7; 299 → 284 for Dio; 153 → 123 for HyTyr; 137 → 137 for Tyr; 539 → 307 and 539 → 275 for Olp; 285 → 133 for Lut; 623 → 161 and 623 → 461 for Ver; 447 → 285 for Lut7 and Lut4; 609 → 301 for Rut; 197 → 121 for Syr. The LC–MS experimental conditions for all the analytes under investigation were as follows: ionspray voltage (IS) -4600 V; curtain gas 23 psi; temperature 400 °C; ion source gas (1) 35 psi; ion source gas (2) 40 psi; collision gas thickness (CAD) medium. The instrument parameters such as entrance potential (EP), declustering potential (DP), collision energy (CE) and cell exit potential (CXP) were optimized for each transition monitored (Table 1). The chromatographic separation was achieved at a flow rate of 300 µL/min with an injection volume of 10 µL. A binary mobile phase made up of 0.1% aqueous formic acid (A) and methanol (B) was gradient programmed to increase B from 5% to 100% in 15 min, hold for 5 min and ramp down to original composition (95% A and 5% B) in five min. Quantitative analyses were performed by external calibration curves built using a least-squares linear regression analysis. For this purpose, standard stock solutions of the phenolic compounds of interest were dissolved in methanol and further diluted with water/0.1% formic acid to obtain calibration standards at concentrations in the range between 100 and 2000 μg/mL. All the solvent used were LC/MS grade; aqueous solutions were prepared using ultrapure water (Millipore, Bedford, MA, USA). The correlation coefficients of the calibration curve ranged between 0.9997 and 0.9999. To evaluate the amount of the oleuropein and ligstroside derivatives, a quantification based on oleuropein glycoside peak areas was deployed. Unfortunately, a direct quantification by means of calibration curves was not possible for these two compound families because they are not commercially available”.

Line 119-127

Table 1 as been add.

Table 1. LC-MS/MS analysis parameters for the phenolic compounds under investigation including molecolar ion [M-H]- monitored on the first quadrupole; main fragment ion selected on the third quadrupole (m/z); declustering potential (DP); entrance potential (EP); collision energy (CE) and cell exit potential (CXP).

Phenolic compound

 [M-H]-1

Main Fragment Ion

DP

EP

CE

CXP

(m/z)

(m/z)

(V)

(V)

(V)

(V)

Catecol

108.8

90.9

-83

-11

-22

-5

Caffeic acid

178.8

135.1

-60

-11

-20

-6

Vanillin

150.9

135.9

-61

-10

-25

-6

Vanillic acid

166.8

108.1

-33

-10

-25

-4

p-Coumaric acid

162.8

119.0

-31

-7

-17

-5

o-Coumaric acid

162.8

119.0

-27

-7

-17

-5

Ferulic acid

192.9

134.1

-35

-10

-21

-5

Apigenin

269.0

117.0

-96

-11

-44

-5

Apigenin-7-O-Glucoside

431.0

267.1

-90

-10

-40

-6

Diosmetin

299.0

284.1

-107

-11

-35

-13

Hydroxytyrosol

152.9

123.0

-80

-11

-22

-5

Tyrosol

136.9

119.1

-60

-8

-20

-6

Oleuropein

539.3

377.1

-65

-10

-20

-5

Luteolin

284.5

133.2

-104

-5

-37

-6

Verbascoside

623.3

161.1

-83

-11

-39

-15

Luteolin-7-O-Glucoside

447.1

285.1

-82

-11

-35

-7

Luteolin-4-O-Glucoside

447.1

285.1

-60

-10

-32

-15

Rutin

609.2

301.0

-116

-11

-50

-15

Syringic acid

196.8

120.9

-70

-11

-25

-6

Line 136

…on oleuropein glycoside peak areas was arranged

now reads:

“…on oleuropein glycoside peak areas was deployed”.

Line 149

Exane was replaced by Hexane

Line 171

Official permission or grant for the animal experiments is missing!

The animal study was reviewed and approved by Ethics Committee for animal testing–CES, of Research Centre for Animal Production and Aquaculture.

Line 177 and 191

The detection was carried out by means of a fluorescence detector, using an excitation wavelength of 297 and an emission wavelength of 328 nm.

The sentence is now:

“The detection was carried out by means of a fluorescence detector, using a range of excitation wavelengths between 295 and 325 nm”.

Line 213-214

DDVOP was produced simply by sucking to destone dried virgin olive pomace, irrespective to pomace chemical extraction. are also produced.

Results and Discussion section was improved and reads:

“The obtaining of this new promising olive industries secondary product, DDVOP, was possible by a simple modification in the flux diagram and technical parameters of traditional technology used to extract pomace oil from virgin olive pomace. The new device differs from the traditional ones for having the dryer set at lower temperature value and for avoiding the solvent extraction phase (Figure 2). In particular, the olive virgin pomace is added from the higher end into a rotary dryer, a cylinder that is slightly inclined from the horizontal direction, mainly composed of a rotating body, a lifting plate, a transmission device, a supporting device and a sealing ring. The high-temperature heat source and the material parallelly flow or counter-flow into the cylinder. As the cylinder rotates, the material runs to the lower end. The lifting board on the inner wall of the cylinder lifts up and drops down the material, so that the contact surface of the material and the air flow is increased to increase the drying rate and promote the advancement of the material. The dried product (Figure 1) exits the dryer from the lower end. The technical characteristics concerning the drying process and the differences between the traditional and the new process in terms of oven and dryer air temperature are shown in Table 2. The new conditions, with the oven outlet air temperature of 350 °C instead of 550 °C, dryer outlet air temperature of 70 °C instead of 80 °C, allow a significant decrease in dried olive pomace temperature, from 70 °C to 40 °C, while leading to a product of the same final moisture (8 %). The innovative flow leading to DDVOP production differs from the traditional ones, also, for having, immediately after the drying device, an aspirator instead an extractor. To produce DDVOP, in fact, no solvent extraction takes place and no pomace oil, to be distilled, is obtained. The aspiration process, instead, allows the removal of the stone making DDVOP suitable as supplement amendment for animal feed……...

Line 219

9,9 % DM?

g/100g dry matter (DM)

Conclusions have been added:

“The technical characteristics concerning the drying process, with the oven outlet air temperature of 350 °C instead of 550 °C, dryer outlet air temperature of 70 °C instead of 80 °C, allow a significant decrease in dried olive pomace temperature, from 70 °C to 40 °C, while leading to a product of the same final moisture (8%). Lowering the temperature of 30 °C allowed to obtain a DDVOP rich in raw protein, precious fiber, phenols, tocopherols, oleic and linoleic acids, hence, making this new product suitable as a new feeding integrator for animal feed. DDVOP, included in a specially formulated feed in a dose of 6%, gave a significant effect (P< 0.05) on milks obtained from sheep fed with DDVOP-enriched feed in terms of phenols, tocopherols, polyunsaturated and unsaturated fatty acids with possible beneficial effect on human health. The production of DDVOP could represent a fine and cheap source of bioactive compounds that positively affect economic status of olive companies while reducing the negative impact of olive by-products on the environment”.

Round 2

Reviewer 1 Report

Authors improved manuscript in accordance with reviewer suggestions. Manuscipt has been enriched with Conclusions, more presice results discussion and better abstract. In my opinion the paper can be accepted. 

Author Response

Thank You!

Reviewer 2 Report

The current manuscript investigates the possibilities of utilizing a by-product rich in selected secondary plant metabolites from the olive industry for animal feeding. The purpose is clearly praxis driven and enables the valorisation of the olive by products. In the following I will try to indicate some parts where I think corrections are recommended.

Line 21-22: Please replace comma with point e.g. 72.29% etc. here and elsewhere in the manuscript:

e.g.

line 261

line 262-3

line 269-270

line 274-277

line 280-282

Line 252: Please replace “zootechnical feeding” with animal feed here

Line 253: …. in simple stomaches are … Be scientifically more precise!

Line 269: … in DDVOP were found to be significantly high. ???

Line 271: These phenolic compounds are well known for their ….

Line 272: … effects derived health benefits and for their anti-inflammatory ….

Line 273 …. properties against atherosclerosis, ….

Line 275: … catechol ...

Line 275-276: p-cumaric/o-cumaric acids – Please do not use capitals and “o” or “p” should be cursive.

Line 278: … tyrosol are known for protecting….

Line 279: …reducing …

Line 279: … was, also, found to be rich …

Line 283: … possess relevant …

Line 254: As the results obtained clearly demonstrated that DDVOP was suitable for a direct use in formulated feed thanks?, ... This is not completely true, since no further toxicological evaluation was done! Please re-write!

Line 292: …. the control group (M1) ?

Line 292: …. with a company concentrate… – What do you mean?

Line 308: … allows a significant …

Tables 3-6: Please replace comma with point for the given values

Author Response

We thank the Reviewer for giving us further inputs to improve the manuscript.

Following are the detailed corrections listed point by point.

The current manuscript investigates the possibilities of utilizing a by-product rich in selected secondary plant metabolites from the olive industry for animal feeding. The purpose is clearly praxis driven and enables the valorisation of the olive by products. In the following I will try to indicate some parts where I think corrections are recommended.

Line 21-22: Please replace comma with point e.g. 72.29% etc. here and elsewhere in the manuscript:

e.g.line 26, line 262-3, line 269-270, line 274-277, line 280-282

Commas were replaced with dots throughout the manuscript (see blue highlighter)

Line 252: Please replace “zootechnical feeding” with animal feed here

Done

Line 253: …. in simple stomaches are … Be scientifically more precise!

Done.

The text now reads: “In fact, fibers, that cannot be used as energy source in monogastric organism having a single-chambered stomach (one stomach) are instead a very good energy source in ruminants as they can degradate them by means of rumen microbes”.

Line 269: … in DDVOP were found to be significantly high. ???

Done

The text now reads: “ ….their concentration was also found to be high in DDVOP…”.

Line 271: These phenolic compounds are well known for their ….

Done

Line 272: … effects derived health benefits and for their anti-inflammatory ….

Done

Line 273 …. properties against atherosclerosis, ….

Done

Line 275: … catechol ...

Done

Line 275-276: p-cumaric/o-cumaric acids – Please do not use capitals and “o” or “p” should be cursive.

Done

Line 278: … tyrosol are known for protecting….

Done

Line 279: …reducing …

Done

Line 279: … was, also, found to be rich …

Done

Line 283: … possess relevant …

Done

Line 254: As the results obtained clearly demonstrated that DDVOP was suitable for a direct use in formulated feed thanks?, ... This is not completely true, since no further toxicological evaluation was done! Please re-write!

Done

The text now reads: “……DDVOP suitable, after further toxicological evaluation, as supplement amendment for animal feed

Line 292: …. the control group (M1) ?

Line 292: …. with a company concentrate… – What do you mean?

Done

The text now reads: “In addition to the basic hay, both groups formed of ten sheep each, the control group (M1) and the experimental one (M2), were fed with a designed specifically for lactating sheep, weather-resistant, protein block with fat, vitamins and minerals added to balance nutrient deficiencies in fair quality forages. Animal feed for M2 was enriched with 6% of DDVOP.”.

Line 308: … allows a significant …

Done

Tables 3-6: Please replace comma with point for the given values.

Done

Commas were replaced with dots (see blue highlighter).